# CrossQuant: A Post-Training Quantization Method with Smaller Quantization Kernel for Precise Large Lanugage Model Compression

## Abstract

Post-Training Quantization (PTQ) is an effective technique for compressing Large Language Models (LLMs). While many studies focus on quantizing both weights and activations, it is still a challenge to maintain the accuracy of LLM after activating quantization. To investigate the primary cause, we extend the concept of kernel from linear algebra to quantization functions to define a new term, "quantization kernel", which refers to the set of elements in activations that are quantized to zero. Through quantitative analysis of the quantization kernel, we find that these elements are crucial for maintaining the accuracy of quantized LLMs. With the decrease of quantization kernel, the precision of quantized LLMs increases. If the quantization kernel proportion is kept below 19% for OPT models and below 1% for LLaMA models, the precision loss from quantizing activations to INT8 becomes negligible. Motivated by the goal of developing a quantization method with small quantization kernel, we propose **CrossQuant**—a simple yet effective method for quantizing activations. CrossQuant cross-quantizes elements using row and column-wise absolute maximum vectors, achieving a quantization kernel of approximately 16% for OPT models and less than 0.1% for LLaMA models. Experimental results on LLMs (LLaMA, OPT) ranging from 6.7B to 70B parameters demonstrate that CrossQuant improves or maintains perplexity and accuracy in language modeling, zero-shot, and few-shot tasks.

## 1 Introduction

In recent years, Large Language Models (LLMs) based on the Transformer architecture (Vaswani et al., 2017) have achieved remarkable success across various domains (He et al., 2024; Dubey et al., 2024; GLM et al., 2024), with model sizes reaching billions and even tens of billions of parameters. However, these LLMs require substantial computational resources for inference. For instance, running the LLaMA3-70B (Dubey et al., 2024) model demands at leas t 140GB of RAM in high-precision (FP16). As the size of LLMs continues to grow, reducing the computational resources required for LLMs inference has become a critical challenge. Quantization, a compression technique, addresses this by reducing model parameters from high-precision floating points (FP16) to low-precision integers (e.g., 8-bit integers, INT8), significantly reducing GPU requirements. To effectively scale larger models on a limited number of devices, it is essential to quantize both weights and activations while utilizing the fewest possible bits, all without compromising accuracy.

Post-Training Quantization (PTQ) compresses LLMs directly without the need for retraining, and can be further divided into two subgroups based on whether activations are quantized: weight-only quantization (Lin et al., 2024; Frantar et al., 2022; Kim et al., 2023) and weight-activation quantization (Xiao et al., 2023; Shao et al., 2024; Yao et al., 2022). There are two widely used methods for quantizing both weights and activations: Per-channel quantization (Liu et al., 2023) and Per-token quantization (Yao et al., 2022). Per-channel quantization has demonstrated superior performance for quantizing weights to INT4/INT8, denoted as W4/W8. Per-token quantization is used in activations. However, Per-token quantization results in significant accuracy degradation when quantizing activations to INT8, denoted as A8. Many studies argue that Per-token quantization

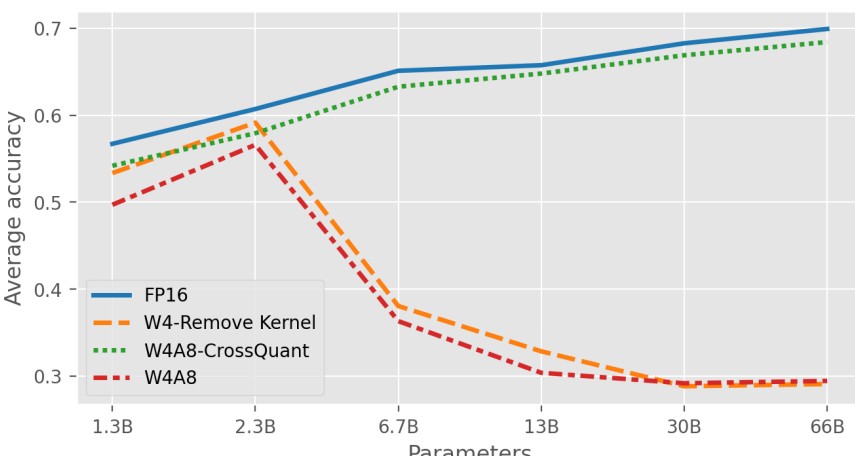

Figure 1: To examine the impact of quantization kernel on the quantization loss, we evaluate the average accuracy of various quantization methods for OPT family models across several zero-shot tasks, including Lambada, ARC-easy, Hellaswag, PIQA, and BoolQ. FP16 serves as the baseline, while W4 refers to weights quantized to INT4. A8 represents activations quantized to INT8, and "Remove Kernel" refers to directly setting the elements in quantization kernel to zero without quantizing the other elements in the activations.

fails to address the challenges posed by outliers in the activation matrix (Kovaleva et al., 2021; Gao et al., 2019; Puccetti et al., 2022), and focus on mitigating the accuracy loss caused by these outliers during quantization (Wei et al., 2022; Dettmers et al., 2022; Yao et al., 2022). However, these methods are still based on the basic idea of Per-token quantization, and cannot achieve better performance in the extreme quantization of lower bits, such as INT4.

**Enhancing quantizaiton precision from the perspective of quantization kernel**. Through quantitative analysis of the accuracy degradation caused by activation quantization, we find that the primary issue stems from the treatment of small-valued elements in activations. Specifically, some elements with small but non-zero absolute values are quantized to zero, forming the quantization kernel. By comparing the effects of setting the elements in quantization kernel to zero with quantizing activations to INT8 (A8), we demonstrate that setting elements in the quantization kernel to zero achieves nearly the same precision as A8. This suggests that most of the error in quantizing activations arises from the quantization kernel (see Figure 1). We find that the precision of quantized LLMs improves with the decrease of the proportion of quantized kernel, as shown in Figure 6 and Figure 7. Furthermore, preserving quantization kernels below a threshold allows quantized models to nearly match the accuracy of FP16, with OPT models at around 19% and LLaMA models at approximately 1%. Thus, the key to achieving high-precision activation quantization is minimizing the kernel. Clear definition and analysis of quantization kernel is provided in section 4.

**Method.** Based on the motivation of finding a quantization method with smaller quantization kernel, we propose CrossQuant, a simple yet powerful approach for quantizing activations to INT8/INT4. Per-token quantization utilizes per-row absolute-maximum vector (denoted as $\boldsymbol{t}$) to quantize the entire activation, as illustrated in Figure 2. When $t_i$ is too large, many elements $X_{i,j}$ in the activation are rounded to zero after division by $t_i$, forming the quantization kernel. CrossQuant introduces the per-column absolute-maximum vector (denoted as $\boldsymbol{c}$) to cross-quantize activations. Since the absolute maximum values of columns are typically smaller than those of rows (see Table 1), most $t_i^{\alpha} c_j^{1-\alpha}$ are smaller than $t_i$ (where $0 \leq \alpha \leq 1$), Consequently, many elements are no longer rounded to zero after division $t_i^{\alpha} c_j^{1-\alpha}$, effectively reducing the quantization kernel, thus preserving precision. Figure 1 demonstrates that CrossQuant applied to INT8 achieves nearly the same accuracy as FP16, and Figure 4 shows the proportion of quantization kernel of CrossQuant and Per-token quantization.

**Experimental results demonstrate the effectiveness of CrossQuant.** In section 5, we apply CrossQuant on the LLaMA family models (Touvron et al., 2023a;b; Dubey et al., 2024) and OPT family models (Zhang et al., 2022), evaluating its performance on language modeling, zero-shot

| Method | Per-token Quantization | CrossQuant (ours) |
|---|---|---|
| Illustration | $t_i$ $X_{i,j}$ $X \in \mathbb{R}^{T \times I}$ $t$ $t = \max(\lvert X_{i,:} \rvert)$ | $c_j^{1-\alpha}$ $c$ $t_i^\alpha$ $X_{i,j}$ $X \in \mathbb{R}^{T \times I}$ $t$ $c = \max(\lvert X_{:,j} \rvert)$ |
| Formula | $Q(X_{i,j}) = round\left(\dfrac{2^{N-1}-1}{t_i} X_{i,j}\right)$ | $CQ(X_{i,j}) = round\left(\dfrac{2^{N-1}-1}{t_i^\alpha c_j^{1-\alpha}} X_{i,j}\right)$ |
| Time complexity | $O(TI)$ | $O(TI)$ |

Figure 2: A comparison table of Per-token quantization and CrossQuant.

and few-shot tasks. Compared to the baselines, CrossQuant shows comparable or superior accuracy across W8A8, W4A8, W4A8-g128 and W4A4.

**Contributions**. (1) We identify that the elements in the quantization kernel mapped to zero are the root cause of the decline in the precision of quantized LLMs. Additionally, we establish the thresholds that OPT and LLaMA models need to stay below to minimize this precision loss. (2) We propose CrossQuant, which determines the scaling factor directly from the activation matrix without requiring additional training. This method maintains the quantization kernel below the identified thresholds to achieve high-precision quantization. (3) CrossQuant demonstrates improved perplexity and accuracy across various model sizes (ranging from 6.7B to 70B) in different model families (LLaMA, OPT).

## 2 RELATED WORKS

Post-Training Quantization (PTQ) of LLM can be divided into two categories: weight-only quantization and weight-activation quantization, based on whether activations are quantized or not (Zhu et al., 2024).

**Weight-only Quantization,** is a method only quantize weights into low-bit integers like INT3 or INT4 with keeping activations in FP16, denoted as W3A16 or W4A16. GPTQ (Frantar et al., 2022) quantizes each channel through iterative refinement, concurrently optimizing the unquantified weights to mitigate and compensate for the difference introduced by the quantization process. AWQ (Lin et al., 2024) focuses on identifying and protecting a small fraction of salient weights based on activation importance. SqueezeLLM (Kim et al., 2023) preserves sensitive weights through a sensitivity-based non-uniform quantization and Dense-and-Sparse decomposition. On devices with limited computing resources, such as mobile devices, quantizing weights alone is insufficient; activation also needs to be quantized.

**Weight-activation Quantization,** quantizes both weights and activations into low-bit values, such as W8A8 (INT8 for both weights and activations). ZeroQuant (Yao et al., 2022) employs both group-wise and token-wise quantization strategies to quantize weights and activations to INT8, marking it as the first deployment of a weight-activation quantization method. Most subsequent works attribute the decrease in quantization accuracy of activations to outliers (Wei et al. (2022); Zou et al. (2024)), which have large-magnitude values and emerge in the activations of models with over 6.7B parameters. LLM.int8() (Dettmers et al., 2022) utilizes a mixed-precision quantization strategy, maintaining outliers and their corresponding weights at FP16, while quantizing the remaining weights and activations to INT8. Outlier Suppression (Wei et al., 2023) and OmniQuant (Shao et al., 2024) focus on identifying and reducing the influence of outlier values in activations

to enhance quantization compatibility. SmoothQuant (Xiao et al., 2023) addresses the ouliers by equivalently transferring the quantization challenge from activations to weights. We find that the quantization kernel is the direct cause of quantization loss, while outliers indirectly affect the size of the quantization kernel (discussed in detail in Appendix A). Our work also aligns with the scope of weight-activation quantization.

## 3 BACKGROUND

Given a linear layer in Transformers (Vaswani et al., 2017), it computes $\boldsymbol{Y} = \boldsymbol{X} \cdot \boldsymbol{W}$, where $\boldsymbol{X} \in \mathbb{R}^{T \times I}$ and $\boldsymbol{W} \in \mathbb{R}^{I \times O}$. $\{T, I, O\}$ indicates {number of tokens, input channels, output channels} respectively. Initial $\boldsymbol{X}$ and $\boldsymbol{W}$ are composed of elements of FP16, and the quantized $Q(\boldsymbol{X})$ and $Q(\boldsymbol{W})$ are composed of elements of INT4 or INT8.

Quantizing activations optimizes model performance by lowering the precision of activation values, which accelerates inference for real-time applications. This reduction decreases the memory foot-print required for activations, improving computational efficiency (Shen et al., 2023). Quantizing weights also minimize memory usage, allowing larger models to fit within the constraints of edge devices, and decreases data transfer, further reducing latency and energy consumption (Lin et al., 2024). Together, weight and activation quantization enable the deployment of sophisticated models in resource-constrained environments, enhancing the practicality of LLMs applications.

**Per-token quantization,** the quantization unit of it is one row and it linearly maps $\boldsymbol{X}$ to integers within the range $[-2^{N-1} - 1, 2^{N-1} - 1]$, which can be represented as:

$$Q(X_{i,j}) = round(\frac{X_{i,j}}{\Delta_{i,j}^x}), \quad \Delta_{i,j}^x = \frac{t_i}{2^{N-1} - 1} = \frac{\max(|\boldsymbol{X}_{i,:}|)}{2^{N-1} - 1}, \quad \forall X_{i,j} \in \boldsymbol{X} \tag{1}$$

where $\Delta_{i,j}^x$ is the element with the maximum absolute value $t_i = \max(|\boldsymbol{X}_{i,:}|)$ in the $i$-th row of $\boldsymbol{X}$. Since the absolute value of outliers is large (at least 20x larger than the other elements), $t_i$ is a large value, resulting in many $X_{i,j}$ are being rounded to zero after dividing by $\Delta_{i,j}^x$, resulting in a large number of elements in the quantization kernel. Our CrossQuant reduces the quantization kernel by reducing $\Delta_{i,j}^x$ so that $\boldsymbol{X}/\Delta_{i,j}^x$ is no longer zero after $round()$.

**Per-channel quantization** is a widely used method for quantizing weights, utilizing the maximum absolute value in the $i$-th row of $\boldsymbol{W}$ to do quantization:

$$Q(W_{i,j}) = round(\frac{W_{i,j}}{\Delta_{i,j}^w}), \quad \Delta_{i,j}^w = \frac{\max(|\boldsymbol{W}_{i,:}|)}{2^{N-1} - 1}, \quad \forall W_{i,j} \in \boldsymbol{W} \tag{2}$$

It is also worth introducing that group-wise quantization is a widely used technique for quantizing weight, which uses smaller channels to achieve higher precision (Shen et al. (2020);Yao et al. (2022);Lin et al. (2024)). It reshapes $\boldsymbol{W} \in \mathbb{R}^{I \times O}$ to $\hat{\boldsymbol{W}} \in \mathbb{R}^{\frac{I \cdot O}{g} \times g}$ first and then does quantization. Many of our experiments are based on W4A8-g128 (group size $g$ is 128) because we want to provide a reference for these weight-only quantization methods using group-wise.

## 4 METHOD

In section 4.1, we give the definition of the quantization kernel. We propose CrossQuant in section 4.2 and demonstrate that CrossQuant has smaller quantization kernel compared to Per-token quantization. The remainder of our findings is presented in section 4.3.

### 4.1 QUANTIZATION KERNEL

In linear algebra, the kernel of a linear map is the part of the domain which is mapped to the zero vector of the co-domain. In this paper, we extend kernel to quantization function to help us define the elements quantized to zero (see Figure 3 for a example of quantization kernel).

**Definition 1.** *Given the quantization method $Q$ and the activation matrix $\boldsymbol{X}$, consider the set of elements quantized to zero, namely the quantization kernel $\mathcal{K}(Q)$:*

$$\mathcal{K}(Q) = \{X_{i,j} \in \boldsymbol{X} \mid Q(X_{i,j}) = 0\} \tag{3}$$

$\mathcal{K}(Q) = \{0.09, -0.1, 0.15, 0.07, -0.2, 0.01, 0.02\}$        $\mathcal{K}(CQ) = \{0.01\}$

| 0 | 127 | 0 | 4 | 4 |
|---|---|---|---|---|
| 0 | 127 | 1 | 0 | 6 |
| 0 | 127 | 2 | 0 | 6 |
| 0 | 127 | 0 | 1 | 3 |

$Q(X)$ (acc 29.24%)

Per-token INT8 →

| 0.09 | 43.4 | -0.1 | 1.4 | 1.2 |
|---|---|---|---|---|
| 0.15 | 58.7 | 0.5 | 0.07 | 2.7 |
| -0.2 | 68.3 | 1.1 | 0.02 | 3.2 |
| 0.01 | 54.8 | 0.2 | 0.5 | 1.5 |

$X$ (acc 69.92%)

CrossQuant INT8 →

| 26 | 86 | -7 | 76 | 32 |
|---|---|---|---|---|
| 41 | 112 | 32 | 4 | 69 |
| -53 | 127 | 68 | 1 | 80 |
| 0 | 105 | 13 | 26 | 39 |

$CQ(X)$ (acc 69.74%)

Figure 3: An example illustrates quantization kernel of two methods on a sample activation matrix $X$, where "acc" is the average accuracy of OPT-66B on five zero-shots tasks: Lambada, ARC-easy, Hellaswag, PIQA and BoolQ.

*Meanwhile, the elements in $\mathcal{K}(Q)$ satisfy:*

$$X_{i,j} \in \mathcal{K}(Q) \Leftrightarrow round(\frac{X_{i,j}}{\Delta_{i,j}^x}) = 0 \Leftrightarrow \left|\frac{X_{i,j}}{\Delta_{i,j}^x}\right| < 0.5 \Leftrightarrow |X_{i,j}| < B_{i,j} \tag{4}$$

*we name $B_{i,j} = 0.5 \times \Delta_{i,j}^x$ as zero bound ($\Delta_{i,j}^x \geq 0$).*

## 4.2 CROSSQUANT

The CrossQuant function can be expressed as following:

$$CQ(X_{i,j}) = round(\frac{X_{i,j}}{\widetilde{\Delta}_{i,j}^x}), \quad \widetilde{\Delta}_{i,j}^x = \frac{t_i^\alpha c_j^{1-\alpha}}{2^{N-1}-1}, \quad \forall X_{i,j} \in X \tag{5}$$

where $c_j = \max(|X_{:,j}|)$ is the maximum absolute value in the $j$-th column. The hyperparameter $\alpha$ within the range $0 \leq \alpha \leq 1$, serves as the exponent component in both $t_i$ and $c_j$, $1-\alpha$ is utilized to maintain normalization. The relationship between the variation of $\alpha$ and the accuracy of the model is discussed in section 5.

Compared to Per-channel quantization, we introduce a vector of "column maximum values" alongside the vector of "row maximum values", collectively cross-quantizing activations. In fact, the quantization unit of CrossQuant is a single element (each element has a different $\widetilde{\Delta}_{i,j}^x$), but does not significantly increase the storage cost with the large increase in quantization accuracy. CrossQuant only stores one extra vector of length $I$ compared with Per-token quantization. About time complexity, $X_{i,j}$ requires one extra division compared to Per-token quantization, but the time complexity is still $O(TI)$.

According to the Definition 1, the kernel of $CQ$ is $\mathcal{K}(CQ) = \{X_{i,j} \in X \mid CQ(X_{i,j}) = 0\} = \left\{X_{i,j} \in X \mid X_{i,j} < \widetilde{B}_{i,j}\right\}$, where $\widetilde{B}_{i,j} = 0.5 \times \widetilde{\Delta}_{i,j}^x$. The results of CrossQuant in this section are all based on $\alpha = 0.15$. We calculate the proportion of kernels caused by two methods relative to the total number of elements in the activation matrix (see Figure 4). For OPT family models with parameters $\geq 2.3B$, the proportion of Per-token quantization kernels experiences a sharp increase (from 16% to 35%) and remains high (between 40% and 55%). In contrast, CrossQuant consistently maintains a low percentage (around 16%). For LLaMA family models, the proportion of Per-token quantization kernels remains low, at approximately 11%, with CrossQuant kernels representing a negligible proportion (<0.1%).

There are a lot of statistical data in Figure 4, but it cannot reflect the corresponding relationship between the quantization kernel of different percentages and the accuracy of the models, so we raise the question: *What is the correlation between quantization kernels of different proportions and the accuracy of quantized models?* To answer this question, we test OPT and LLaMA models on language modeling task of different settings, shown in Figure 5, Figure 6 and Figure 7.

From the combination of Figure 4 and Figure 5, we observe the following results: (1) Generally, the size of quantization kernels is positively correlated with perplexity; (2) Quantization kernels larger

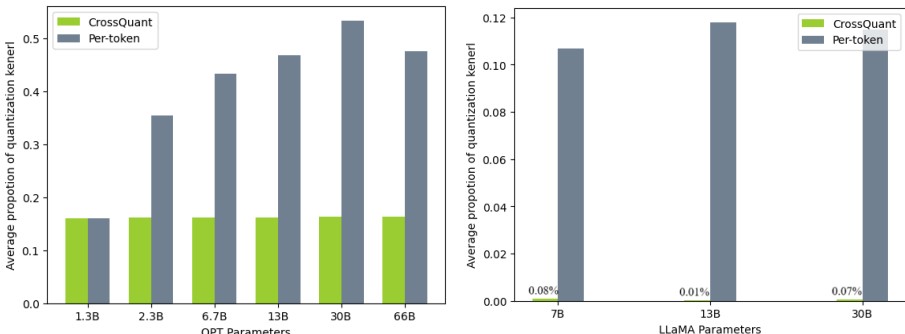

Figure 4: The average proportion of kernels of both quantization methods are calculated in all activations in OPT (left) and LLaMA (right) models on WikiText2.

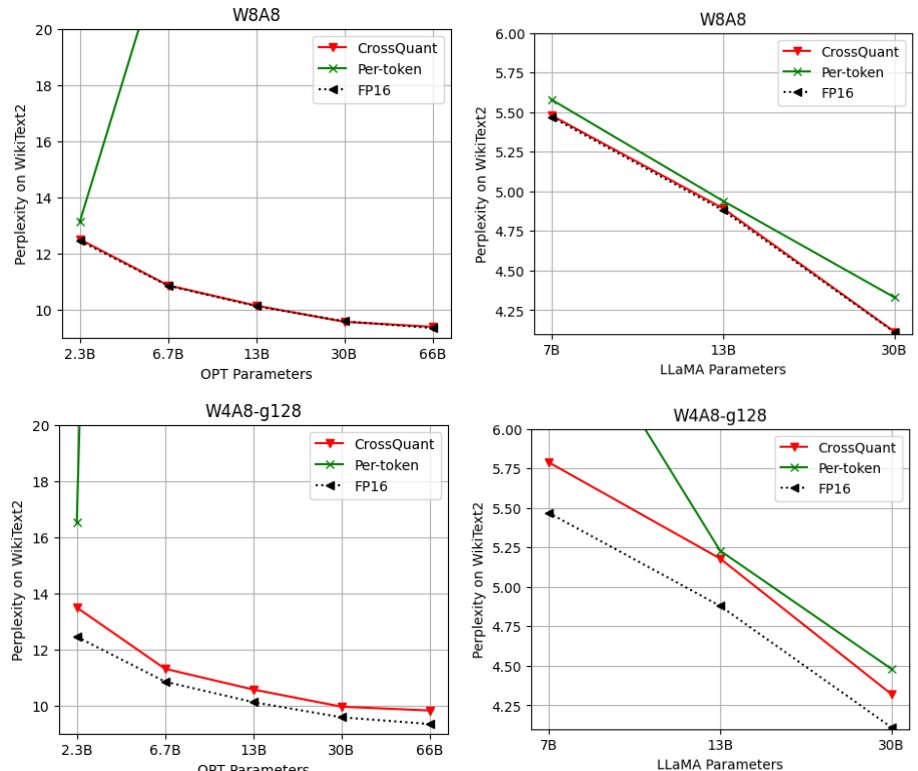

Figure 5: We evaluate the OPT and LLaMA models on the language modeling task using the Wiki-Text2 dataset, measuring performance via perplexity. The top two groups are tested with W8A8, while the bottom two groups use W4A8-g128.

than 40% lead to a significant decline in the perplexity of quantized models, as seen in the figures of the OPT models; (3) CrossQuant outperforms Per-token quantization due to its smaller quantization kernel; (4) Different models exhibit different thresholds, below which the proportion of quantized kernels mitigates the accuracy loss caused by activations. Through experiments, we find that this threshold is approximately 19% for OPT models and 1% for LLaMA models, and we discuss about these thresholds in detail in section 4.3.

We now provide a theoretical analysis showing that $\mathcal{K}(CQ)$ is smaller than $\mathcal{K}(Q)$. Given the activation matrix $\boldsymbol{X}$, the elements in $\boldsymbol{X}$ are invariant. Therefore, proving $\mathcal{K}(CQ)$ is smaller than $\mathcal{K}(Q)$ only depends on proving $\widetilde{B}_{i,j} < B_{i,j}$. For convenience, we write $\max(|\boldsymbol{X}_{i,:}|)$ as $t_i$, $\max(|\boldsymbol{X}_{:,j}|)$ as $c_j$. There are two cases for $c_j$ and $t_i$ are discussed:

I: $\quad c_j < t_i$

Scaling the original formula as $t_i^\alpha c_j^{1-\alpha} < t_i$, from $\widetilde{\Delta}_{i,j}^x < \Delta_{i,j}^x$ we can know that $\widetilde{B}_{i,j} < B_{i,j}$.

II:   $c_j \geq t_i$

Case II will lead to $\widetilde{B}_{i,j} \geq B_{i,j}$, but it actually takes a small proportion (about 3%) in the whole matrix, as shown in Table 1. When $\alpha = 0.75$, the proportion of $\widetilde{B}_{i,j} < B_{i,j}$ is the highest, but the result is not the best, because the average proportion of quantization kernel does not the lowest.

Table 1: The average proportion of OPT-13B in activations for the tow indicators are calculated on WikiText2. $\alpha = 1$ is actually Per-token quantization, thus $\widetilde{B}_{i,j} < B_{i,j}$ is not counted.

| OPT-13B | $\alpha = 0.15$ | $\alpha = 0.45$ | $\alpha = 0.75$ | $\alpha = 1$ |
|---|---|---|---|---|
| $c_j \geq t_i$ | 3.10% | 3.11% | 2.76% | 0.93% |
| $\widetilde{B}_{i,j} < B_{i,j}$ | 96.84% | 96.82% | 97.14% | - |
| Quantization Kernel | 16.17% | 16.22% | 16.32% | 43.40% |
| W8A8 perplexity | 10.13 | 10.20 | 10.83 | 3e+4 |

### 4.3 DETERMINE THE THRESHOLD OF THE QUANTIZATION KERNEL

In section 4.2, we establish a positive correlation between different proportions of the quantization kernel and model accuracy. In this subsection, we quantitatively analyze the relationship between the proportion of quantization kernel and the model accuracy, while also exploring the specific threshold values for LLaMA and OPT models mentioned in section 4.2 for LLaMA models and OPT models. As illustrated in Figure 6 and Figure 7, the thresholds for OPT-6.7B, 13B, 30B, and 66B are 25%, 19%, 25%, and 20%, respectively; for LLaMA2-7B, 2-13B, and 1-30B, the thresholds are 2%, 1%, and 1%. Therefore, the quantization kernel should be reduced to below 19% for OPT models and 1% for LLaMA models to achieve activation quantization without precision loss.

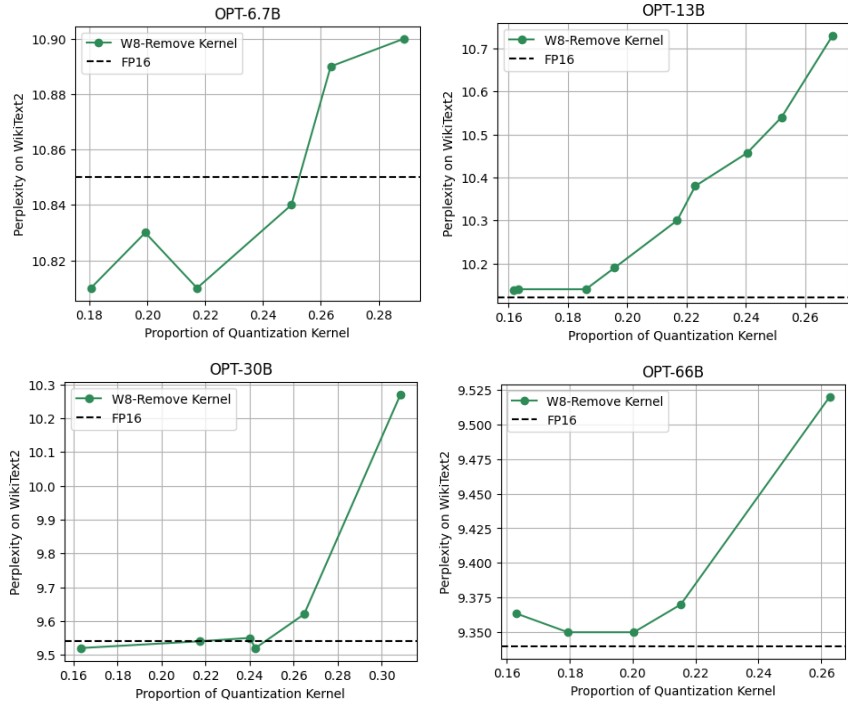

Figure 6: Perplexity of OPT models on the language modeling task using the WikiText2 dataset, where "W8-Remove Kernel" indicates quantizing weights to INT8 and setting different proportion of quantization kernels to zero directly without quantizing other elements in activations.

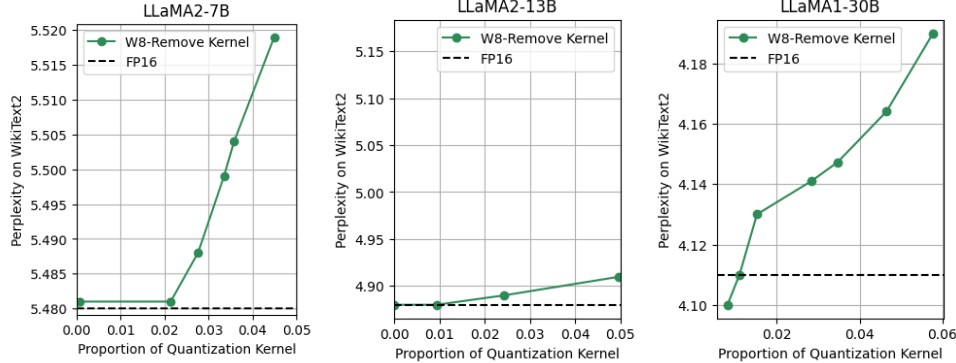

Figure 7: Perplexity of LLaMA models on the language modeling task using the WikiText2 dataset, where "W8-Remove Kernel" indicates quantizing weights to INT8 and setting different proportion of quantization kernels to zero directly without quantizing other elements in activations.

## 5 EXPERIMENTS

### 5.1 SETUPS

**Models**. We implemented CrossQuant on the LLaMA family models (2-7B, 2-13B, 1-30B, 3-8B, 3-70B) (Touvron et al., 2023a;b; Dubey et al., 2024) and the OPT family models (1.3B, 2.3B, 6.7B, 13B, 30B, 66B) (Zhang et al., 2022).

**Baselines**. Both weight-activation quantization and weight-only quantization are chose as baselines. For weight-activate quantization, we compare CrossQuant with Per-token quantization, SmoothQuant (Xiao et al., 2023) and OmniQuant (Shao et al., 2024). For weight-only quantization, we select AWQ (Lin et al., 2024) with activations quantized by Per-token quantization. The weights quantizaiton method for CrossQuant is Per-channel quantization.

**Evaluation**. Quantized models are evaluated on language modeling experiments, zero-shot and few-shot tasks. Language modeling experiments include WikiText2 (Merity et al., 2016) and C4 (Raffel et al., 2019); zero-shot tasks include Lambada (Paperno et al., 2016), ARC-easy (Clark et al., 2018), PIQA (Lourie et al., 2021), Hellaswag (Zellers et al., 2019) and BoolQ (Clark et al., 2019); the few-shot task is MMLU (Hendrycks et al., 2021) with five-shots.

Table 2: Perplexity(↓) results of quantized LLaMA models (2-7B, 2-13B, 1-30B) on language modeling tasks in three groups: W8A8, W4A8-g128 and W4A4. The datasets are the test split of Wikitext2 and C4.

| Method | W/A | 7B | | 13B | | 30B | |
| --- | --- | --- | --- | --- | --- | --- | --- |
| | | Wiki2 | C4 | Wiki2 | C4 | Wiki2 | C4 |
| FP16 | W16A16 | 5.47 | 7.52 | 4.88 | 7.01 | 4.11 | 6.38 |
| Per-token | W8A8 | 5.58 | 7.69 | 4.94 | 7.71 | 4.33 | 7.66 |
| SmoothQuant | W8A8 | 5.51 | 7.58 | 4.92 | 7.03 | 4.20 | 6.50 |
| CrossQuant | W8A8 | **5.48** | **7.53** | **4.89** | **7.00** | **4.11** | **6.42** |
| Per-token | W4A8-g128 | 6.99 | 8.07 | 5.23 | 7.24 | 4.48 | 8.05. |
| AWQ | W4A8-g128 | 5.79 | 7.92 | 5.14 | 7.19 | 4.39 | 6.64 |
| CrossQuant | W4A8-g128 | 5.79 | **7.81** | 5.18 | 7.18 | 4.32 | 6.61 |
| CrossQuant+AWQ | W4A8-g128 | **5.70** | **7.81** | **5.14** | **7.15** | **4.27** | **6.51** |
| Per-token | W4A4 | 2e+4 | 2e+4 | 5e+4 | 4e+4 | 1e+4 | 1e+4 |
| OmniQuant | W4A4 | 13.0 | 18.89 | 14.2 | 18.0 | 9.15 | 14.32 |
| CrossQuant | W4A4 | **12.40** | **18.19** | **7.59** | **7.85** | **7.48** | **10.69** |

## 5.2 LANGUAGE MODELING TASKS

Table 2 contains three groups of experiments to test the perplexity of CrossQuant on language modeling task. Meanwhile, we check the combination effect of CrossQuant with AWQ. For the first group (W8A8), CrossQuant demonstrates slightly better performance on the 7B and 13B models compared to SmoothQuant. In the second group, CrossQuant matches the performance of AWQ, with 4 wins and 2 losses. Notably, CrossQuant+AWQ achieves improved perplexity, indicating that CrossQuant can be effectively combined with weight-only methods for better results. In the last group, CrossQuant reduced perplexity by 4.8%-56.38% compared to OmniQuant.

## 5.3 ZERO-SHOT TASKS

The all results in Table 3 are obtained by the `lm-eval-harness`[1]. CrossQuant closely matches the FP16 accuracy on both W8A8 and W4A8-g128, performing slightly better than SmoothQuant on W8A8. For W4A8-g128 and W4A8, CrossQuant significantly outperforms the baselines. We also test CrossQuant on OPT-1.3B, 2.3B, 6.7B and 13B, shown in Appendix B.2. Overall, CrossQuant demonstrates its abilities to maintain the accuracy across quantized LLMs of varying sizes.

Table 3: Accuracy($\uparrow$) results of OPT models (30B, 66B) on five zero-shot tasks, including Lambada, ARC-easy, PIQA, Hellaswag and BoolQ. Average accuracy of each baseline is calculated at the last column.

| Model | Method | W/A | Lambada | ARC-easy | PIQA | Hellaswag | BoolQ | Avg. |
|---|---|---|---|---|---|---|---|---|
| OPT-30B | FP16 | W16A16 | 70.63% | 68.86% | 77.52% | 54.25% | 70.12% | 68.27% |
| | Per-token | W8A8 | 0.00% | 29.62% | 55.16% | 26.42% | 37.85% | 29.81% |
| | SmoothQuant | W8A8 | **71.41%** | 69.57% | 77.42% | 54.22% | 69.94% | 68.51% |
| | CrossQuant | W8A8 | 70.77% | **69.90%** | **77.80%** | **54.32%** | 70.09% | **68.57%** |
| | Per-token | W4A8-g128 | 0.00% | 27.02% | 53.21% | 26.13% | 37.82% | 28.36% |
| | AWQ | W4A8-g128 | 0.01% | 25.79% | 53.31% | 27.33% | 42.26% | 29.74% |
| | CrossQuant | W4A8-g128 | **68.26%** | **68.35%** | **76.98%** | **53.16%** | **67.37%** | **66.82%** |
| | Per-token | W4A4 | 0.00% | 26.39% | 51.85% | 25.93% | 37.83% | 28.40% |
| | OmniQuant | W4A4 | 0.00% | 24.4% | 51.5% | 25.88% | 37.85% | 27.92% |
| | CrossQuant | W4A4 | **43.12%** | **58.50%** | **69.70%** | **39.89%** | **61.74%** | **54.59%** |
| OPT-66B | FP16 | W16A16 | 73.58 | 71.63% | 78.72% | 56.36% | 69.35% | 69.92% |
| | Per-token | W8A8 | 0.00% | 28.95% | 53.42% | 26.00% | 37.85% | 29.24% |
| | SmoothQuant | W8A8 | 73.1% | 70.79% | 78.45% | 56.21% | 67.77% | 69.26% |
| | CrossQuant | W8A8 | **73.61%** | **71.38%** | **78.67%** | **56.34%** | 68.72% | **69.74%** |
| | Per-token | W4A8-g128 | 0.00% | 28.03% | 53.80% | 25.77% | 37.85% | 29.09% |
| | AWQ | W4A8-g128 | 1.94% | 25.79% | 53.31% | 27.33% | 42.26% | 30.12% |
| | CrossQuant | W4A8-g128 | **72.46%** | **68.68%** | **77.20%** | **54.49%** | **69.23%** | **68.41%** |
| | Per-token | W4A4 | 0.00% | 24.66% | 50.97% | 26.02% | 37.82% | 27.89% |
| | OmniQuant | W4A4 | 0.00% | 24.87% | 51.25% | 25.89% | 37.82% | 27.96% |
| | CrossQuant | W4A4 | **26.21%** | **54.58%** | **62.51%** | **34.33%** | **51.59%** | **45.84%** |

## 5.4 ABLATION STUDY

In ablation study, we mainly explore the influence of different values of $\alpha$ on perplexity and accuracy. As shown in Figure 8, on the Lambada dataset, OPT-6.7B has a qualitative leap in accuracy (from 43% up to 80%) and reaches the optimal value at $\alpha = 0.55$. On WikiText2, perplexity drops significantly (from 6.99 to bleow 5.09) after $\alpha \leq 0.95$, and the optimal value is obtained at $\alpha = 0.15$.

Then we test CrossQuant's performance on LLaMA3-8B and 3-70B, see Table 4. CrossQuant has a wide but narrow lead over SmoothQuant on $\alpha = 0.15$, a partial advantage on $\alpha = 0.45$ and $\alpha = 0.75$. Ablation study shows that CrossQuant performs well with $\alpha \leq 0.55$ in general. The closer $\alpha$ is to 1 (the closer it is to Per-token quantization), the worse LLMs performs.

---

[1]https://github.com/EleutherAI/lm-evaluation-harness

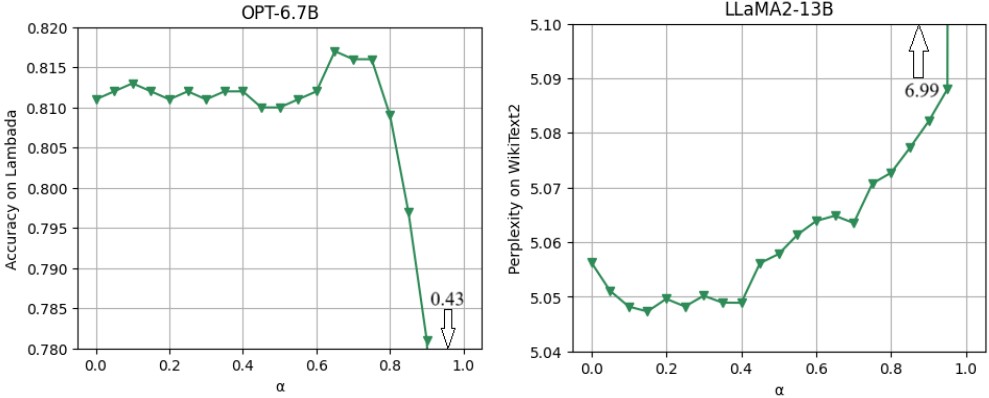

Figure 8: Visualizations of accuracy changes for OPT-6.7B-W8A8 on Lambada (left) and perplexity changes for LLaMA2-13B-W4A8 on WikiText2 (right) with varying $\alpha$.

Table 4: Ablation study results on LLaMA3-8B and 3-70B

| Method | W/A | llama3-8B | | llama3-70B | |
|---|---|---|---|---|---|
| | | Wiki2 | MMLU | Wiki2 | MMLU |
| FP16 | W16A16 | 6.13 | 65.25% | 2.85 | 78.90% |
| Per-token | W8A8 | 6.27 | 64.40% | 41.90 | 28.99% |
| SmoothQuant | W8A8 | 6.25 | 64.40% | 2.97 | 78.39% |
| CrossQuant $\alpha = 0.15$ | W8A8 | **6.16** | **65.40%** | **2.93** | **78.57%** |
| CrossQuant $\alpha = 0.45$ | W8A8 | 6.17 | 65.30% | 2.94 | 78.33% |
| CrossQuant $\alpha = 0.75$ | W8A8 | 6.20 | 64.94% | 3.23 | 74.57% |

## 6 LIMITATIONS AND FUTURE WORK

Our work has the following limitations, which are also our future research directions:

- Although we have identified that the decrease in quantization accuracy is caused by quantization kernels, this conclusion is based on experimental results. We have not yet fully explored the specific information contained in undershoots that necessitates their preservation during quantization. In future work, we will continue to investigate the missing features during quantization inference.

- The OPT-2.3B model exhibits a high proportion of kernels yet still performs well after quantization (see Figure 4). We hypothesize that this is related to the scale of the model's parameters. In future research, we will examine why smaller models can tolerate a high percentage (30%) of quantization kernels.

## 7 CONCLUSION

In this paper, we introduce the concept of "quantization kernel", defined as the set of elements in activations that are quantized to zero. Investigating the primary reason for the decline in precision in quantized LLMs, we find that this decline is attributable to the quantization kernel being mapped to zero. Through quantitative experiments, we establish that the quantization kernels need to be reduced to a certain proportion to maintain the accuracy of quantized LLMs, specifically 19% for OPT models and 1% for LLaMA models. CrossQuant, a weight-activation method designed to minimize the quantization kernels, outperforms the baselines in language modeling, zero-shot, and few-shot tasks, with its quantization kernels of 16% for OPT models and <1% for LLaMA models. Our method provides a new analytical perspective to better understand and deconstruct quantization loss, and we hope our findings inspire further valuable research in the field of quantization.

## REPRODUCIBILITY STATEMENT

We guarantee that all results presented in this paper are reproducible. To ensure the reproducibility of our results, we provide the following details regarding our methodology and data: for data availability, all datasets used in this paper are publicly accessible at `https://huggingface.co/`; for code availability, a source code is provided in supplementary materials; for experimental setup, detailed experiments settings could be seen in Appendix B.1.

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

## A    OUTLIERS IN TRANSFORMER

Sparse but systematically large-magnitude outliers (only 0.1% of all input features but are at least 20x larger than the other values (Dettmers et al., 2022)) significantly emerge in activations of Large Language Models (LLMs) with over 6.7B parameters. Many previous works contribute the decline of quantized models accuracy to outliers. We think that the decrease of quantization accuracy caused by outliers is mainly due to the excessive quantization kernel. As shown in Per-token quantization function:

$$Q(X_{i,j}) = round(\frac{X_{i,j}}{\Delta_{i,j}^x}), \quad \Delta_{i,j}^x = \frac{t_i}{2^{N-1}-1} = \frac{\max(|\boldsymbol{X}_{i,:}|)}{2^{N-1}-1}, \quad \forall X_{i,j} \in \boldsymbol{X} \qquad (6)$$

outliers make the $t_i$ become large, and $\boldsymbol{X}_{i,j}$ is a small value which would be rounded to zero after dividing by $t_i$. Thus outlier leads to the large size of quantization kernel. To avoid this problem, the direct way is let $\Delta_{i,j}^x$ be smaller, so our CrossQuant is proposed around this idea. There are also many previous studies have examined the causes of outliers and their relationship to perfor-mance degradation in quantized models (Gao et al., 2019; Timkey & van Schijndel, 2021; Dettmers et al., 2022). Kovaleva et al. (2021) found that the emergence of outliers in the BERT model family is linked to the normalization process of LayerNorm. Additionally, Puccetti et al. (2022) demon-strated through experiments that the appearance of outliers is related to the frequency of tokens in the training distribution. Devlin et al. (2019) introduced novel quantization schemes, such as

Per-embedding-group Quantization for BERT, which addresses the issue of quantized models disproportionately focusing on special tokens.

The existing works have studied outliers in detail, and we analyze quantization loss from quantization kernel.

# B    SUPPLEMENTARY MATERIAL FOR EXPERIMENTS

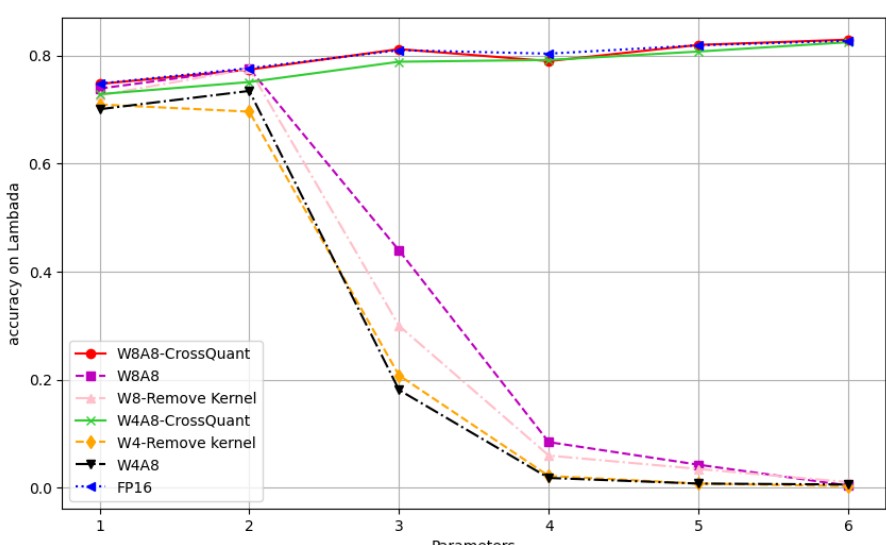

Figure 9: To examine the impact of quantization kernel on the quantization loss of activations, we evaluate the average accuracy of various quantization methods for OPT family models across several zero-shot tasks, including Lambada, ARC-easy, Hellaswag, PIQA, and BoolQ. FP16 serves as the baseline, while W4/W8 refers to weights quantized to INT4/INT8, and A8 represents that activations are quantized to INT8, and "Remove Kernel" refers to directly setting the elements in kernel to zero without quantizing the other elements in the activations.

As shown in Figure 9, the accuracy of W4 and W8, A8 and "Remove Kernel" are comparable, indicating that the quantization kernel plays a major role in the accuracy degradation of Large Language Models (LLMs). CrossQuant achieves similar results to FP16 on both W4 and W8.

## B.1    EXPERIMENTS SETTINGS

We deploy all of our experiments on RTX 4090 except OmniQuant. For each baseline, the specifics of our implementation are:

**SmoothQuant**. We generate smooth scales from SmoothQuang's open source code[2] and use its $fake\_quant.py$ to get all results. The smooth factor $\alpha$ for LLaMA and OPT is 0.8 and 0.5 respectively (follow the settings in its demo).

**AWQ**. We generate AWQ search results by its open source code[3]. Since AWQ is a weight-only method, we deploy fake quant (Per-token quantizaiton) on activations during inference. And AWQ's code only supports group-wise quatization for weights with size 128 and 64, we choose g128, that is W4A8-g128. For W4A4, CrossQuant shows significant decline but are still better than baselines.

**OmniQuant**. OmniQuant is a weight-activation method but needs extra training to generate OmniQuant parameters. We test both our own generated OmniQuant parameters and OpenGVLab's

---

[2]https://github.com/mit-han-lab/smoothquant
[3]https://github.com/mit-han-lab/llm-awq

open source OmniQuant parameters[4], and each model selected better results as the baseline. All of OmniQuant's experiments are deployed on the NVIDIA A800.

**CrossQuant**. We use Per-channel quantization to quantize weights in CrossQuant except LLaMA3-70B and OPT-66B. To deploy experiments on our method, we still need to determine the value of $\alpha$. As discussed in section 4, we find that $\alpha = 0.15$ is a general good value for all models, and may not be the best. All experiments of CrossQuant are implemented with $\alpha = 0.15$.

When we quantizing OPT-66B to W4A4 and LLaMA3-70B to W8A8, we meet with great resistance, due to the kernels of Per-channel quantization in weights matrix. Some works have found that outliers also emerge in weights of models (Dettmers et al., 2023; Kim et al., 2023), which will cause kernels in weights quantized by Per-channel quantization. CrossQuant as a solution solving the problem of quantization kernels, it can be used in weights as well. Thus we deploy CrossQuant on both weights and activations on quantizing OPT-66B to W4A4 and LLaMA3-70B to W8A8. Given $\alpha_X = 0.15$, we find optimal $\alpha_W$ for OPT-66B and LLaMA3-70B, that is $\alpha_W = 0.55$ and $\alpha_W = 0$ respectively.

Our experimental results can be reproduced based on the following open source code AWQ[5] and SmoothQuant[6] by adding the following fake quant code in the appropriate place:

```
def CrossQuant(x, n_bits=8, alpha=0.15):
    q_max = 2 ** (n_bits - 1) - 1
    scale_t = x.abs().max(dim=-1, keepdim=True)[0].pow(alpha).div_(q_max)
    scale_c = x.abs(dim=-2)[0].pow(1-alpha)
    x.div_(scale_t).div_(scale_c).round_().mul_(scale_c).mul_(scale_t)
    return x
```

From the perspective of the code, CrossQuant only adds a point division operation compared to Per-token quantization, while Per-token quantization itself requires a matrix point division, so CrossQuant's time complexity does not increase.

## B.2 ZERO-SHOT EXPERIMENTS

We supplement CrossQuant's experiments on OPT family models, as shown in Table 5, with data that are also complementary to Figure 1. From the table we can see that Per-token's accuracy drops rapidly when model parameters are $\geq 6.7$B (outliers start to emerge), while crossquant still maintains an accuracy close to that of FP16.

---

[4]https://huggingface.co/ChenMnZ/OmniQuant/tree/main

[5]https://github.com/mit-han-lab/llm-awq

[6]https://github.com/mit-han-lab/smoothquant

Table 5: Accuracy(↑) results of OPT models (1.3B, 2.3B, 6.7B, 13B) on five zero-shot tasks, including Lambada, ARC-easy, PIQA, Hellaswag and BoolQ. Average accuracy of each baseline is calculated at the last column.

| Model | Method | W/A | Lambada | ARC-easy | PIQA | Hellaswag | BoolQ | Avg. |
|---|---|---|---|---|---|---|---|---|
| OPT-1.3B | FP16 | W16A16 | 56.14% | 57.57% | 71.54% | 41.37% | 56.97% | 56.71% |
| | Per-token | W8A8 | 57.07% | 57.25% | 70.62% | 40.76% | 57.25% | 56.29% |
| | CrossQuant | W8A8 | 56.39% | 56.94% | 71.27% | 41.33% | 56.45% | 56.47% |
| | Per-token | W4A8-g128 | 52.04% | 54.16% | 70.07% | 39.61% | 50.88% | 53.35% |
| | CrossQuant | W4A8-g128 | 53.79% | 56.14% | 70.29% | 40.27% | 50.48% | 54.19% |
| OPT-2.3B | FP16 | W16A16 | 63.46% | 60.73% | 73.78% | 45.91% | 59.69% | 60.71% |
| | Per-token | W8A8 | 66.66% | 57.74% | 72.85% | 44.19% | 60.21% | 60.33% |
| | CrossQuant | W8A8 | 63.47% | 60.94% | 74.37% | 45.83% | 60.48% | 61.01% |
| | Per-token | W4A8-g128 | 60.78% | 56.31% | 72.47% | 42.91% | 57.21% | 57.93% |
| | CrossQuant | W4A8-g128 | 61.01% | 59.42% | 73.12% | 44.99% | 57.21% | 59.15% |
| OPT-6.7B | FP16 | W16A16 | 67.30% | 65.61% | 76.22% | 50.53% | 65.93% | 65.11% |
| | Per-token | W8A8 | 17.64% | 47.26% | 62.02% | 36.97% | 60.42% | 44.86% |
| | CrossQuant | W8A8 | 67.22% | 65.45% | 76.50% | 50.49% | 65.63% | 65.05% |
| | Per-token | W4A8-g128 | 2.98% | 38.13% | 57.72% | 31.70% | 59.81% | 38.06% |
| | CrossQuant | W4A8-g128 | 63.36% | 65.02% | 75.62% | 49.08% | 63.36% | 63.28% |
| OPT-13B | FP16 | W16A16 | 68.15% | 67.17% | 75.79% | 52.39% | 65.26% | 65.75% |
| | Per-token | W8A8 | 0.00% | 27.94% | 53.64% | 26.39% | 55.04% | 32.60% |
| | CrossQuant | W8A8 | 68.04% | 66.96% | 76.33% | 52.40% | 65.14% | 65.77% |
| | Per-token | W4A8-g128 | 0.00% | 25.96% | 52.06% | 26.29% | 59.96% | 32.85% |
| | CrossQuant | W4A8-g128 | 66.56% | 66.45% | 75.89% | 50.67% | 64.40% | 64.79% |

