# OpenReview forum: "CrossQuant: A Post-Training Quantization Method with Smaller Quantization Kernel for Precise Large Lanugage Model Compression"
_ICLR.cc/2025/Conference — ICLR 2025 Conference Withdrawn Submission_

### Official Review · Reviewer_xqTQ · 2024-10-28

**Soundness:** 2
**Presentation:** 3
**Contribution:** 2
**Rating:** 5
**Confidence:** 4

**Summary:**

This paper discovered the performance decline caused by weight-activation quantization is the quantization kernal, a novel concept  presented by the authors that representing the zero elements after quantization. To address the issue, CrossQuant is proposed to successfully reduce the number of quantization kernal and increase the quantized model performance.

**Strengths:**

1. To the best of our knowledge, this paper is the first to demonstrate the influence of zero elements after quantization on model performance and give a fresh perspective except for the well-known outliers.
2. In some experimental settings, CrossQuant achieves almost lossless level compared with FP16 LLM.

**Weaknesses:**

1. CrossQuant is not a hardware-friendly method. Compared with traditional per-token activation quantization, CrossQuant utilizes additional column-wise scaling strategy (Eq.5) to reduce the number of quantization kernals. Per-channel (column-wise) scaling does not map well to GEMM kernals, because when there are multiple sets of scaling factors in one row of activation tensor, it is not conducive to parallel acceleration. In generel, scaling can only be performed along the outer dimensions of the matrix multiplication.
2. The authors ignore to analysis the reason why OPT and LLaMA show quite distinctions when calculating the proportions of quantization kernals, which may enhance the depth of the paper.

**Questions:**

1. For weakness1, the best solution for exploiting the advantages of CrossQuant is to design specific acceleration kernal. And I know this is not easy to achieve during rebuttal, so the authors should give the latency/throughput on real system compared with other per-token weight-activation PTQ method (i.e., SmoothQuant) to disprove my thought.
2. Lack of detailed experimental setup, such as calibration sets and hardware usage.
3. Please maintain consistency of the case of model name in tables (i.e., llama3-8b in Table 4 and LLaMA models in Table2).

---

### Official Review · Reviewer_ERXu · 2024-10-29

**Soundness:** 2
**Presentation:** 2
**Contribution:** 1
**Rating:** 3
**Confidence:** 5

**Summary:**

This paper proposes the CrossQuant quantization method, aimed at quantizing large language models. Specifically, the authors introduce a column-wise max factor to reduce the dimensionality of the ”quantization kernel”. A series of experiments demonstrate the effectiveness of CrossQuant.

**Strengths:**

1. This paper introduces a new factor that effectively reduces the proportion of values quantized to zero.
2. Experiments show that the proposed method effectively reduces the perplexity of large language models (LLMs).

**Weaknesses:**

1. According to the description in SmoothQuant, it is not advisable to calculate the quantization scale in the iner dimension of tensors, such as the input_channel dimension of weights and activations, as this can lead to dequantization issues. Therefore, a question arises: can the column-wise max introduced in CrossQuant accelerate computation during quantization? In fact, the original per-token quantization for $X$ combined with per-channel quantization for $W$ can represent the quantized multiplication in the following form:
$$Y={\delta}^{-1}X_{int8}W_{int8}{\gamma}^{-1}.$$
Where $X_{int8}W_{int8}$ represents the accelerated low-precision matrix multiplication, and ${\delta}^{-1}{\gamma}^{-1}$ are the dequantization scales. However, how should the dequantization formula be expressed after introducing the column-wise max factor in CrossQuant? Can low-precision matrix multiplication still be achieved to obtain quantization benefits? The paper does not discuss the efficiency gains from quantization.
2. Some new methods have established a new benchmark for LLM quantization, particularly in the W4A4 quantization configuration. Examples include AffineQuant[1], QuaRot[2], and SpinQuant[3]. As far as I know, these methods were open-sourced three months before the ICLR submission deadline, thus complying with ICLR's citation rules.
3. Experiments based on a series of new models should be included in the paper. Models like OPT mentioned in the paper are somewhat outdated, and experiments with newer models such as LLaMA3 and QWEN should be incorporated.


[1] AffineQuant: Affine Transformation Quantization for Large Language Models. ICLR 2024.
[2] QuaRot: Outlier-Free 4-Bit Inference in Rotated LLMs. Arxiv 2024.
[3] SpinQuant: LLM quantization with learned rotations. Arxiv 2024.

**Questions:**

1. The issues mentioned in the weaknesses should be addressed.
2. Is there a correlation between the size of $B_{ij}$ and the size of the quantized kernel? According to the results in Table 1 of the paper, $\widetilde{B}_{ij}<B_{ij}$, but the quantized kernel actually increases in size. Therefore, the validity of the hypothesis in L321 is questionable.
3. A personal opinion: the term “quantization kernel” defined in the paper (even though this concept might be referring to the “kernel” in mathematics) can easily cause confusion with the "kernel" in quantization CUDA programming. Perhaps it could be renamed to "null space" or something else?
4. Spelling and grammar errors:
a) L83, L407, L749, “quantization”->”quantization”
b) L181, “of it” -> “of which”

---

### Official Review · Reviewer_Gsqz · 2024-10-30

**Soundness:** 3
**Presentation:** 2
**Contribution:** 2
**Rating:** 3
**Confidence:** 3

**Summary:**

This paper presents a concept called Quantization kernel for analyzing the effect of activated quantization on the accuracy of LLMs and finds that maintaining a low quantization kernel ratio is critical to maintaining model accuracy. Based on this finding, the researchers developed the CrossQuant method. Experimental results show that CrossQuant improves, or at least maintains, the PPL and accuracy of LLMs (such as LLaMA and OPT) on multiple tasks.

**Strengths:**

1. The method seems technically sound and straightforward in principle.
2. Good written and easy to follow.

**Weaknesses:**

1. Rescaling activation is not a particularly new thing. The author lacks the comparison of several methods, such as QuaRot and AffineQuant. Please note that their accuracy are better than OmniQuant and SmoothQuant.
2. The author uses an exponent less than 1 to calculate scale, will there be additional calculation delay? What has happened to the inference speed of the method compared to the baseline (like BF16 and SmoothQuant/OmniQuant)? It is better for the author to give detailed experimental results. For example, the TTFT (Time To First Token), TPOT (Time Per Output Token) and Throughput (token/second for all parallel requests) are evaluated with different batch sizes (e.g. 1,2,4,8,16,32) and sequence lengths (e.g. 64,128,256,512,1024,2048).
3. Although previous work (such as OmniQuant/SmoothQuant) quantified the OPT model. However, as an LLM proposed in 2022, the accuracy of OPT is not high. For example, the accuracy of OPT on MMLU is only 25-30%, which is close to a random guess. Strongly advise the authors to focus on LLaMA2/3.
4. However, the authors only give the PPL of LLaMA2 on C4/W2 in Table 2, which is actually a little strange. Because PPL does not reflect the true performances of the LLM itself. For Table 3, its evaluation is too simplistic. It is suggested that the author evaluate the MMLU(-pro) with 0/5-shot, IFEval, GSM8K, HumanEval and other datasets according to Table 2 in the llama3 technical report [3].

[1]: QuaRot: Outlier-Free 4-Bit Inference in Rotated LLMs, In NIPS24
[2]: AffineQuant: Affine Transformation Quantization for Large Language Models, In ICLR24
[3]: The Llama 3 Herd of Models. In ArXiv.

**Questions:**

Please see the weakness. I hope the author can add further experimental details, including, but not limited to, comparisons with other methods, actual speed measurements, and more test results, to prove the effectiveness of the method.

---

### Official Review · Reviewer_3dxK · 2024-10-31

**Soundness:** 2
**Presentation:** 3
**Contribution:** 2
**Rating:** 3
**Confidence:** 5

**Summary:**

This paper introduces a novel perspective on post-training quantization for Large Language Models (LLMs) through the concept of "quantization kernel". To address the challenge of accuracy preservation in quantized LLMs, the authors propose CrossQuant, an efficient quantization method that leverages both row and column-wise absolute maximum vectors from the activation matrix to determine scaling factors. This approach stands in contrast to conventional per-token quantization methods, which solely utilize row-wise maximum values. By incorporating this dual-directional scaling strategy, CrossQuant effectively reduces the quantization kernel size, thus preserving a larger proportion of non-zero elements during the compression process. The method's efficacy is empirically validated through comprehensive experiments across multiple architectural families.

**Strengths:**

1. The paper demonstrates excellent organization and clarity, with well-structured presentation of the problem, methodology, and comprehensive experimental results.
2. The paper presents a novel theoretical contribution through the introduction of "quantization kernel," supported by rigorous quantitative analysis. The authors demonstrate how their proposed CrossQuant methodology effectively mitigates accuracy degradation by optimizing the quantization kernel dimensions, offering a principled approach to LLM quantization.
3. CrossQuant is both elegant in its simplicity and powerful in its effectiveness, requiring no additional training while being compatible with various LLM quantization approaches across different model families and sizes.

**Weaknesses:**

1.The experimental evaluation lacks comprehensive representation. The experiments primarily focus on select sizes of LLaMA-1 and LLaMA-2 models, notably missing results from the more recent LLaMA-3 and LLaMA-3.1 series. The comparison baseline is also insufficient, as the paper could include benchmarking against newer quantization methods such as Atom[1], SpinQuant[3] and AffineQuant[4]. Furthermore, the ablation studies on α parameter selection are limited to a single model series, which is inadequate to demonstrate the method's generalizability across different architectures.
2.While the paper demonstrates empirically that cross-quantizing elements using row and column-wise absolute maximum vectors effectively reduces the quantization kernel, the introduction of this specific approach appears abrupt and lacks clear motivational reasoning. The authors primarily rely on experimental results to validate their method without explaining the intuition or thought process behind choosing this particular solution.
3.The paper presents limited hardware performance analysis. Despite being a quantization method, there is insufficient discussion of practical deployment metrics such as actual inference speedup, memory consumption, and hardware utilization across different computing platforms. These real-world performance measurements would better validate the method's practical utility.

**Questions:**

Given the demonstrated success of integrating CrossQuant with AWQ, I would be particularly interested in understanding whether and how the authors envision extending this integration to more recent quantization approaches such as Quarot[2] or SpinQuant[3], as such combinations might further advance the state-of-the-art in LLM quantization.

[1] Atom: Low-bit Quantization for Efficient and Accurate LLM Serving. MLSys 2024.
[2] QuaRot: Outlier-Free 4-Bit Inference in Rotated LLMs. ArXiv.
[3] SpinQuant: LLM Quantization with Learned Rotations. ArXiv.
[4] AffineQuant: Affine Transformation Quantization for Large Language Models. ICLR 2024.

---

### Note · Authors · 2024-11-25

**Comment:**

We sincerely thank all four reviewers for their thoughtful and detailed evaluations of our work. It is clear that each reviewer carefully read our paper and provided valuable feedback, highlighting its strengths and weaknesses for improvement. After thorough consideration, we have decided to withdraw our submission in order to revise it extensively based on the constructive comments received. Finally, we would like to express our gratitude to the ICLR 2025 conference for providing us with the opportunity to engage with expert reviewers and receive such high-quality feedback, which will greatly benefit the future development of our work.

**Withdrawal Confirmation:**

I have read and agree with the venue's withdrawal policy on behalf of myself and my co-authors.